# The Surface Condition of Ni-Cr after SiC Abrasive Blasting for Applications in Ceramic Restorations

**DOI:** 10.3390/ma13245824

**Published:** 2020-12-21

**Authors:** Weronika Czepułkowska-Pawlak, Emilia Wołowiec-Korecka, Leszek Klimek

**Affiliations:** Institute of Materials Science and Engineering, Faculty of Mechanical Engineering, Lodz University of Technology, Stefanowskiego 1/15, 90-924 Łódź, Poland; emilia.wolowiec-korecka@p.lodz.pl (E.W.-K.); leszek.klimek@p.lodz.pl (L.K.)

**Keywords:** abrasive blasting, Ni-Cr alloy, silicon carbide, surface condition, fixed dental restorations, PFM crowns

## Abstract

Abrasive blasting is a process widely used in dentistry. One of the uses is the development of metal surfaces for connections with ceramics in fixed prosthetic restorations. The purpose of this paper was to check how the rough surface profile (width, height, and depth on unevenness) impacts the surface’s condition, like its wettability and percentage of stuck abrasives. The Ni-Cr alloy surface was abrasive blasted by silicon carbide with the various pressure parameters (0.2, 0.4, and 0.6 MPa) and abrasive particle sizes (50, 110, and 250 µm). Cleaned surfaces were examined for roughness, wettability, and percentage of stuck abrasive particles on the surface. The surface after abrasive blasting using 110 µm of abrasive size and 0.4 MPa pressure has the best wettability results. The width of unevenness may cause it. When the unevenness has too small or too large width and depth, the fluids may not cover the entire cavities because of locking the air. The surface condition of dental alloys directly affects metal–ceramic connection strength. The knowledge about the impact of the abrasive blasting parameters on the bond strength will allow one to create durable dental restorations.

## 1. Introduction

Abrasive blasting, more commonly known as sandblasting, is a process of forcibly propelling the steam of abrasive material on the surface under high pressure, which results in smoothing a rough surface, roughening a smooth surface, removing surface contaminants, or shaping a surface. [1]. The surface’s final appearance is related to the operation of several factors that influence its condition in various ways. The first of these is connected to the properties of the material subject to processing and, in particular, its hardness. Tough materials, under the abrasive influence, will be affected by machining to a small degree, but abrasive particles hitting with a large force may cause microcracks and chipping of the machined material. On the other hand, materials with lower hardness are more susceptible to the effect of the abrasive, which is why their profile is characterized by higher roughness after the processing. Another factor is the type of abrasive used for processing. These materials differ from one another in terms of hardness, density, size, and shape; finally, they can shape the surface being processed differently [2,3]. Moreover, the parameters of the same process influence the final condition of the processed material surface. The speed of particles hitting the material surface influences it to a varying degree, causing smaller or greater changes in its topography. The next factor is related to the processing medium. Whether the process is performed in a dry or wet environment may influence the profile creation in various manners. Wet processing is less aggressive for the surface, and dusting is avoided [4]. Additionally, it is used when it is necessary to add a chemical ingredient to the medium, which dissolves ingredients that appear on surfaces [5,6].

Assessment of the surface’s condition is used to control all production processes and allows for an analysis of functional surface parameters related to the material used. An early analysis of the working conditions of the material, and the selection of surfaces allowing the best possible surface preparation for use, is important mostly in terms of the duration of their use and the prevention of possible damage. In dentistry, analyses of the influence of metal alloy roughness on the bonding with ceramics are performed [7]. Depending on the profile appearance, a ceramic material can enter the rough surface in various ways. The width and depth turn out to be important where indentations that are too shallow and narrow may not form mechanical connections. Indentations that are too wide or too deep, on the other hand, may lock the air inside and reduce the permanence of the filling.

Many factors influence the condition of the surface after abrasive blasting processes, thanks to which it has many applications. One of them is developing the surfaces of dentistry alloys such as the nickel-chromium alloy, cobalt-chromium alloy, or titanium for mechanical bonds with dentistry ceramics. Such fillings are characterized by good durability and are broadly produced in clinical cases [8]. Ni-Cr alloy is widely used in dentistry because of its biocompatibility and good properties, which increase the strength of dental restorations (crowns, bridges, skeletal prosthesis, post, and core). It is characterized by a high modulus of elasticity, good corrosion resistance, and low price. The important feature in the creation of ceramic restorations is their good thermal compatibility with porcelain. However, nickel can cause allergies in contact with tissues. Beryl (Be) is also often added to alloy for better castability, but it is a carcinogenic element. Therefore, novel alloys avoid adding Be to the alloy composition. Despite the disadvantages, Ni-Cr alloy still is widely used in dentistry, as well as in metal–ceramic restorations. One problem is related to the chipping of the ceramic material from the metal base surface [9]. There are studies that analyze the influence of abrasive blasting using its various parameters on the durability of this bond; however, they mostly concern processes using aluminum oxide as an abrasive. There is a cognitive gap for the Ni-Cr alloy and abrasive blasting processes related to silicon carbide as the abrasive. There is not research on Ni-Cr alloy in the literature data, which would concern detailed analysis of roughness parameters tests, other surface properties such as its wettability, and the percentage of stuck abrasive particles, which are related to the impact of variety abrasive blasting parameters, i.e., the influence of different sandblasting pressures and different particle sizes of silicon carbide. The research will check how the rough surface profile (width, height, and depth on unevenness) and the percentage of stuck abrasives impact the surface’s wettability. The impact of the abrasive blasting parameters to the surface unevenness and percentage of stuck abrasive particles will also be analyzed. In previous research, which concerns other alloys (Co-Cr alloy, titanium) and other abrasive (alumina), which parameters of surface roughness after treatment with variable parameters influence its wettability has not been interpreted. Usually, the literature provides basic parameters such as Ra (arithmetical mean deviation of the assessed profile) and Rz (maximum height of the profile), which are general and do not describe the profile’s appearance. The authors do not analyze the issue more broadly. Silica carbide is an abrasive with sharper shapes and is slightly harder than alumina. It can prove to be an excellent alternative to alumina owing to better efficiency in sandblasting the surface. Moreover, the composition includes silicon, which is the main component of some dental ceramics. Therefore, it may prove to be a factor in increasing the bond’s strength when the semi-fluid materials fuse during firing. Therefore, the aim of this study is analyze the condition of the nickel-chromium alloy surface, i.e., its surface, fluid ability to moisten it, and the percentage of stuck abrasive particles after processing using various parameters with silicon carbide abrasive.

## 2. Materials and Methods

Eighteen Heraenium^®^ NA nickel-chromium alloy samples (Heraeus Kulzer, Hanau, Germany) were shaped like cylinders with a diameter of 7 mm and a height of 15 mm. Chemical composition and mechanical properties were presented in Table 1 and Table 2. One of the cylinder bases was blast cleaned (Alox 2001, Effegi Brega, Sarmato, Italy) using silicon carbide (SiC) for 20 s, with a nozzle inclination at 45° and a distance of 15 mm from the surface of the material. All samples were divided into nine groups, each of which contained two samples. The groups were distinguished by abrasive blasting parameters where the abrasive particle size and the processing pressure were the variables (Table 3).

After abrasive blasting, all samples were cleaned in an ultrasonic cleaner (Emmi-55HC-Q, Emag, Poland) in deionized water for eight minutes to remove loose abrasive particles, and then the surface was dried under compressed air.

The samples prepared in the way described above were tested for the following:-surface roughness;-the wettability of the surface;-the percentage of stuck abrasive particles.

The roughness was measured using the Hommel-Etamic T8000 (Jenoptik, Germany) surface roughness machine. A measuring probe with a radius of 0.005 mm was used. The length of profile mapping was 400 µm at a measuring speed of 0.5 mm/s. Three profiles were prepared for each surface. Surface parameters were calculated by the profilometer software according to standard PN-EN ISO 4288:2011 [10]: Ra (arithmetical mean deviation of the assessed profile); Rz (maximum height of the profile); Rt (total height of the profile); Rp (maximum peak height); Rv (maximum valley depth); Rq (root mean squared); Rc (mean height of profile elements); Rsm (mean width of the roughness profile elements); RV_0_ (retention volume); and parameters related to the material ratio: Rk (core roughness depth), Rpk (reduced peak height), Rvk (reduced valley depth), Mr1, and Mr2 (material ratio).

The surface’s wettability was examined using measuring fluids and the FM40 EasyDrop device (Krüss GmbH Germany, Hamburg, Germany). Received wettability angles were used to calculate the surface free energy using the Owens–Wendt method [11].

An examination was performed for the backscattered electron (BSE) imaging to distinguish between phases using a scanning electron microscope. A total of 15 photographs were taken for each sample, which were used to analyze the surface further to determine the percentage of stuck abrasive particles. For this purpose, ImageJ 1.48v (Wayne Rasband, National Institutes for Health, Bethesda, MD, USA) software was used. Each photograph was prepared for calculations by increasing the contrast between the places where the abrasive was present and the metal’s surface. Subsequently, each image was binarized and analyzed for the percentage of anchored particles (Figure 1). Figure 2 presents the scheme of the research carried out.

## 3. Results

The surface roughness measurement results are presented in Table 4 and Table 5 and Figure 3 and Figure 4.

For the roughness parameters presented in Table 4, the results increase with an increase in particle size. The great differences for each particle size can be observed between the lowest pressure used (0.2 MPa) and higher pressure (0.4 and 0.6 MPa). For the latter, the differences in their values are small. The lowest values can be observed for sample S52 and the highest for S24–S26.

Table 5 presents the results related to the material percentage in the sample roughness share. An increase in the value of parameters related to the profile height (Rk, Rpk, and Rvk) can be observed, together with an increase in the particle size used. The opposite situation can be observed for Mr1 and Mr2, that is, a bigger particle size and higher pressure lower parameters.

Figure 3 and Figure 4 present graphs of the material percentages for samples S56 and S24. Grey color on the figures determines the tested material; blue color refers to the area between the depths and the profile core, and yellow defines the amount of material in the highest peaks of the profile.

The highest differences can be observed in the number of peaks related to the profile height, where S56 has the highest number as compared with S24. Their heights are also different. For the first sample, much lower peaks can be observed—confirmed by the peaks contained in Table 5 (Rpk). The same tendency can be observed for the depth of profile indentations (Rvk) and the height of the roughness core (Rk). A large difference is also observed for the width of roughness in both figures—also confirmed by the data in Table 4 (Rsm).

Figure 5 presents the results related to the wetting angles formed after wetting the surface with the measuring fluid (water and diiodomethane). For each surface, higher values of water wetting angles can be observed, and no significant differences in the values of these angles can be observed for diiodomethane. Samples sandblasted with the smallest particle (50 µm) have the highest values of angles related to the surface’s wettability with water; the lowest values, on the other hand, can be observed for samples S14 and S16.

The data related to the surface free energy and its components (polar and dispersive ones) are presented in Figure 6. The highest surface free energy (SFE) value can be observed for surfaces after using the 110 µm abrasive under a pressure of 0.4 MPa. For both these samples and S16, the lowest differences can be observed between the components’ values. The other samples are characterized by a much greater spread between the values of both components. For the S54 sample, where it is the highest, the lowest SFE value can be observed.

Table 6 presents the results concerning the percentage of abrasive particles stuck in the alloy surface. A linear relationship can be observed for the smallest particle used together with increased processing pressure. The highest percentage can be observed for the remaining particle sizes after the processing at a pressure of 0.4 MPa. Similar values can be observed for samples S56, S16, and S22. The lowest results were obtained for sample S52 and the highest for S24.

## 4. Discussion

Surface roughness measurements after abrasive blasting confirm the relationship between the processing parameters and the surface parameters. The use of various pressure values and various abrasive particle sizes results in changes in the processed material. The results show that the lowest pressure (0.2 MPa) causes a decrease in the values of roughness parameters by comparing them to the parameters obtained after processing at higher pressures. The energy of abrasive particles is not high for this pressure; thus, the alloy surface is not strongly formed. The unevenness heights and depths are small, and the roughness core (Rk) is the smallest. The tested surface roughness parameters increase as the abrasive material increases. The 0.4 and 0.6 MPa pressures cause the energy of abrasives to be greater, and as a result, the unevenness is higher and deeper. In Figure 3 and Figure 4, differences between the prepared profiles are visible. These are obvious relationships that are confirmed in numerous studies [3,7,12,13,14]. The influence of roughness on the applications in various areas is broadly examined in the literature. However, the basic parameters of Ra and Rz are mostly used to describe the unevenness. In our study, we focused on a broader surface analysis for the applications in dentistry, i.e., to be more precise, for the creation of ceramic fillings, the topography related to the unevenness width and indentation depth. Amplitude parameters such as Ra or Rz are mostly used to control process data, where surface anomaly detection is important [15]. For dental prosthetic purposes, the profile’s appearance is important, which is why graphs were created in this study to show the material percentage in the profile, which is shown in Figure 3 and Figure 4. Considerable differences can be observed in the width of roughness unevenness, confirmed in the values of the Rsm parameter—which describes this property of the profile (Table 4). The pressure of treatment affects the profile’s height and depth, as the abrasives’ size affects the unevenness width. It follows that, bigger abrasive, bigger Rsm. It is important as it can influence the fluid’s ability to wet the surface and its overflow onto the created unevenness, which may later determine the durability of the bond between the metal base with the outer ceramics. Moreover, the important RV_0_ parameter, which describes the oil capacity, depends on the size of the abrasive and the process pressure used. A larger abrasive size, which hits the metal surface at a high pressure, forms uneven structures with a high depth of indentations (Table 4). Theoretically, this should be advantageous in dental prosthetic applications; however, there is a risk that air will be locked in these indentations while the surface is being wetted with semi-liquid ceramics. Indentations that are too narrow, which result from the impact of small abrasive particles, may also be inappropriate because of a failure to form appropriate mechanical connections to allow the bond with ceramics. In this case, there is also the risk of locking the air in the unevenness owing to their small width. However, the degree of changes results from both the material being processed and the abrasive itself. For example, in tests of the titanium surface condition after sandblasting using aluminum oxide, the values of basing roughness parameters such as Ra and Rz for surfaces after processing using the particle of 110 µm, under a pressure of 0.4 MPa, is 1.029 µm (Ra) and 7.605 µm (Rz) [16]. For the same processing parameters, an Ra of 1.2 µm and Rz of 12.1 µm can be observed for a cobalt-chromium alloy [17]. The surface roughness, after processing, is also influenced by the type of the abrasive used for the processing. It is related to its hardness and particle shape, where rounded abrasive particles form roughness on the surface with a greater width of unevenness and a smaller depth than particles with sharp shapes. A study by Taga et al. compared the influence of using various abrasives on the Co-Cr alloy surface [3]. The measured Ra parameter proved to be the highest for the surface sandblasted with SiC. The lowest roughness was achieved for samples after the processing using glass beads and the mixture. The SEM (Scanning Electron Microscope) photographs’ surface was much smoother as compared with the surface after sandblasting with an abrasive with sharper edges. Similar results were obtained in zirconium oxide [18] and titanium [19] research. By comparing the results obtained in this study with the results from [20]—where the same nickel-chromium alloy was used, but aluminum oxide was the abrasive—it can be observed that the influence of the processing parameters on the surface condition changes as a result of both the pressure and the abrasive size. However, the values of measured parameters are higher after the processing using silicon carbide. In summary, the abrasive used for abrasive blasting influences the results obtained, which is related to both its hardness and shape. Silicon carbide is an abrasive with sharp shapes, which results in greater machining of the processed material, and the surfaces formed turn out to have more developed roughness profiles.

The direct influence of the surface roughness and the described parameters can be observed in examining the fluid ability to wet the Ni-Cr alloy. The optimal processing method proved to be the one using medium-sized silicon carbide (110 µm) at medium pressure (0.4 MPa). Both small and (too) large abrasive particle sizes cause a change in the SFE values (Figure 6), which results from the surface topography and, more specifically, the unevenness width. Small distances between uneven areas made these surfaces more hydrophobic. Rough areas proved to be too narrow or too wide, and most likely, the locking of the air in the uneven spaces results in their lower wettability. A similar situation was achieved for titanium surfaces after sandblasting using aluminum oxide [16]. The blasting using the smallest particle size resulted in the achievement of a more hydrophobic surface. The highest SFE parameters were obtained for the surface after the use of the medium-sized particle. Slightly different results were obtained for the cobalt-chromium alloy. In this case, the highest SFE was achieved for the largest particle, and all surfaces were hydrophobic [17,21]. In our study, the assumption of roughness’s impact on the surface wettability with fluid was confirmed. The use of the smallest and largest particle resulted in higher wetting angles related to the surface roughness.

The last test performed was an examination of the percentage of stuck abrasive particles. The lowest values were observed for the surface after the blasting with the use of the smallest particle. This can be caused by the abrasive bouncing off the metal, which is related to too low particle energy that prevents it from sticking into the alloy. With growing particle size, the percentage increases as particles become stuck in the metal owing to the increased energy. Additionally, they have a greater surface as compared with the smallest particle. The results of this test are also influenced by the hardness of both the alloy and the abrasive. Suppose the aluminum oxide is used as the abrasive, a significant increase in the abrasive amount stuck in the Ni-Cr alloy [22]. A percentage of 18.2% is obtained for the smallest particle and the lowest pressure used in the process. This value increases to 24.8% if the largest particle and the highest pressure are used. This situation can be related to the abrasive hardness where, for silicon carbide (a material with a higher hardness than aluminum oxide), the particle may crack upon hitting the surface [22]. As a result, the abrasive is present in a smaller area of the surface. This applies to each size of the particle. Moreover, a change of the material results in a change of values of the tested surface property. For the cobalt-chromium alloy, an amount of Al_2_O_3_ particle stuck for a surface blasted using the lowest parameters is 16.95% and, for the highest, it is 20.94% [23]. This alloy is a harder material than the Ni-Cr alloy; therefore, its surface is occupied by the abrasive to a lesser extent. The role of abrasive particles stuck in the surface for the bond with dentistry ceramics is unknown. Sharp particle shapes sticking out of the surface may initiate tensions, which ultimately reduce the bond’s durability. It is also possible that particles are chemically bonded to the ceramics and increase the bond’s durability. However, these assumptions should be confirmed in experimental studies.

In summary, the nickel-chromium surface condition after abrasive blasting with the use of silicon carbide is variable depending on the parameters used in the process. Sanding with an abrasive, with a size of 110 µm at 0.4 MPa, may prove the optimal processing. Parameter values can be observed for this surface, which may indicate the formation of a permanent bond with dental ceramic. A semi-liquid ceramic material by overflowing the uneven surface after the blasting—with both the lowest pressure and the smallest particle—may not form a sufficient mechanical connection. The unevenness width and profile indentation depths are too small, and thus the connection durability may be too small. The use of the largest particle and the highest pressure results in large distances between uneven areas. The depth of the indentations may prove to be appropriate. Still, their widths may contribute to reduced durability of the connection, hence the assumption that the surface condition after using 110 µm and 0.4 MPa may prove to be the best for the formation of metal–ceramics bonds. Moreover, the best results related to the fluid’s ability to wet the surface can be observed for these blasting parameters. Experimental research to examine the durability of metal–ceramics bonds will need to be conducted to confirm this assumption.

## 5. Conclusions

The research shows that the surface’s description with the basic roughness parameters (Ra and Rz) is insufficient. In dentistry, the surface characteristics should be described using parameters describing, among others, the width of the roughness (Rsm) and the oil volume RV_o_. They turn out to be important in the surface wetting properties. From the dentistry perspective, the most optimal treatment may be made with the medium grain (110 µm) under the pressure of 0.4 MPa. For these parameters, the best surface wetting properties were achieved. Abrasive blasting also causes the abrasive to stick to the surface. These particles’ role in the metal–ceramic bond’s strength is not yet known, and more extensive research is needed.

## Figures and Tables

**Figure 1 materials-13-05824-f001:**
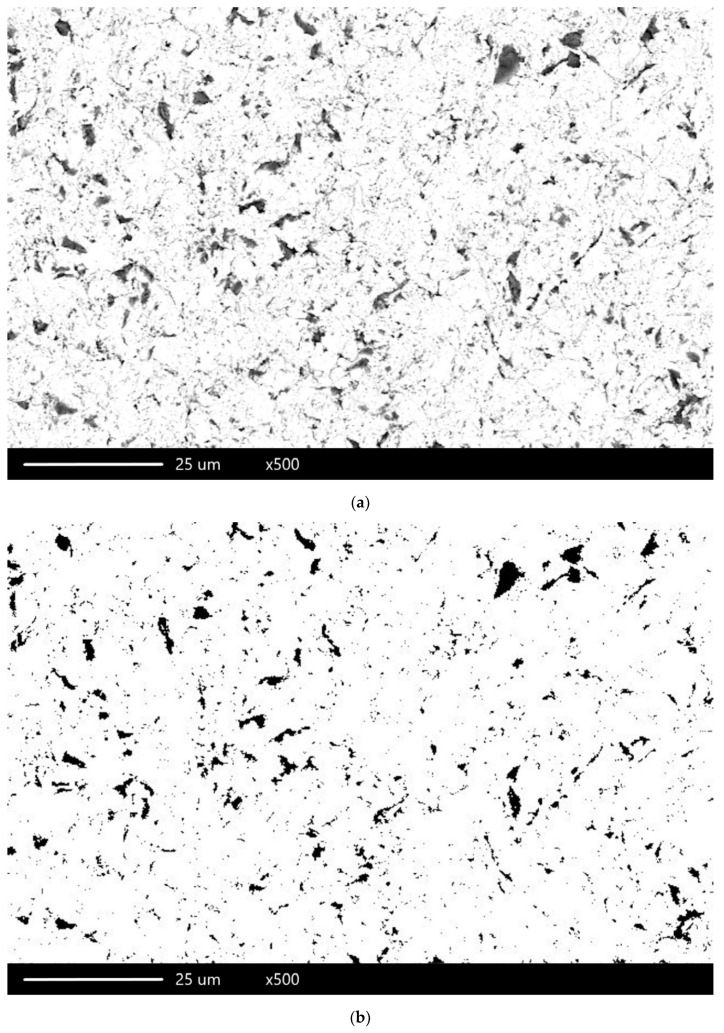
Photograph of the Ni-Cr alloy surface after SiC blasting cleaning size 50 µm, pressure 0.2 MPa: (**a**) the image before the processing in the program; (**b**) the image after it.

**Figure 2 materials-13-05824-f002:**
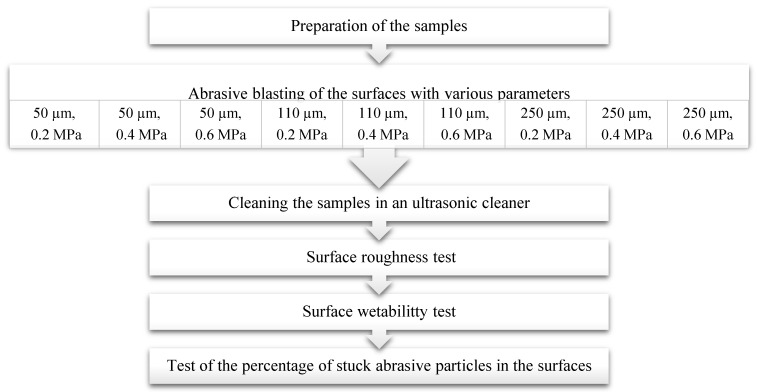
Flow chart presenting the step-wise procedure of the research.

**Figure 3 materials-13-05824-f003:**
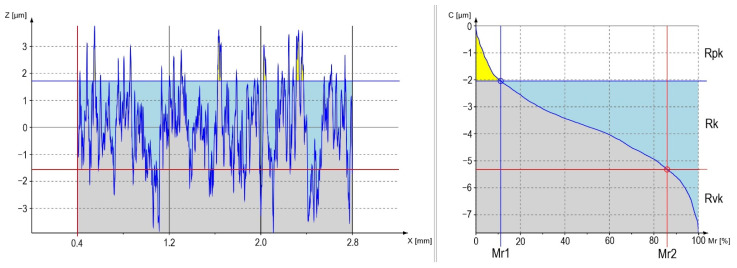
Material ratio curve in the profile of sample S56.

**Figure 4 materials-13-05824-f004:**
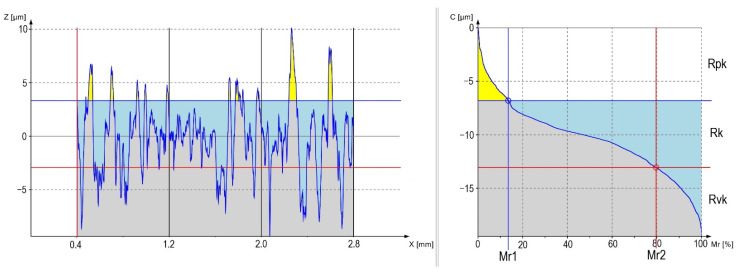
Material ratio curve in the profile of sample S24.

**Figure 5 materials-13-05824-f005:**
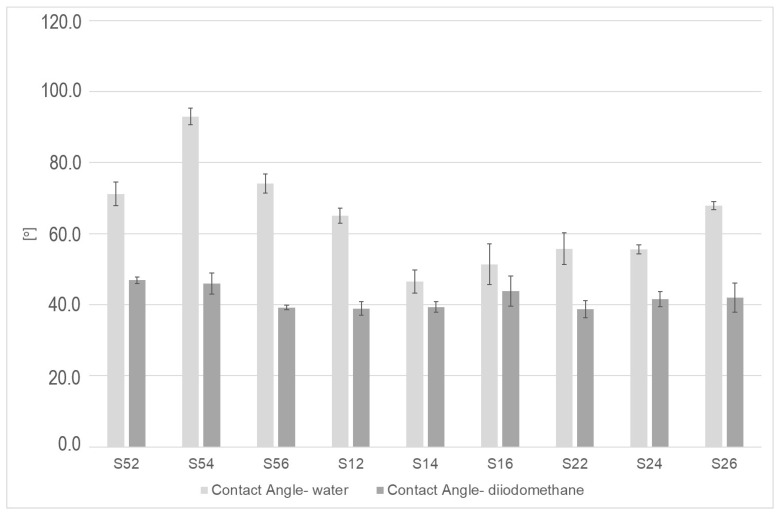
Graph presenting water and diiodomethane wetting for the surface after SiC processing.

**Figure 6 materials-13-05824-f006:**
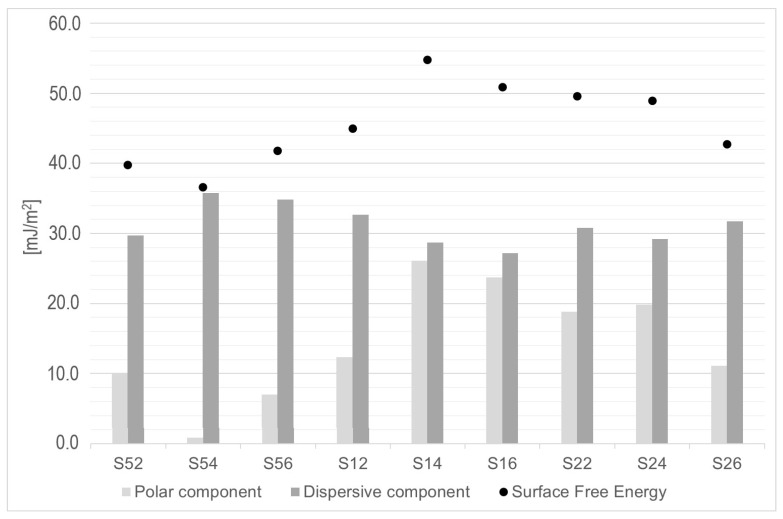
The polar and dispersive components and surface free energy (SFE) for the surface after SiC processing.

**Table 1 materials-13-05824-t001:** Chemical composition of the Heraenium^®^NA alloy (wt.%).

Ni	Cr	Mo	Fe	Mn	Ta	Si	Nb
59.3	24.0	10.0	1.5	1.5	1.5	1.2	1.0

**Table 2 materials-13-05824-t002:** Heraenium^®^NA alloy properties.

Alloy Properties	
Density	8.3 g/cm^3^
Melting temperature	1190–1300 °C
Coefficient of thermal expansion (25–500 °C)	14.1 µm/m × K
Young’s modulus	227 GPa
Yield point	350 MPa
Elongation at break	20%
Hardness [HV 10]	185 HV

**Table 3 materials-13-05824-t003:** The parameters of abrasive blasting processes.

Abrasive Particle Size [µm]	Processing Pressure [MPa]		
	0.2	0.4	0.6
50	S52	S54	S54
110	S12	S14	S16
250	S22	S24	S26

**Table 4 materials-13-05824-t004:** Surface roughness results.

Sample Number	Roughness Parameters								
	Ra [µm]	Rz [µm]	Rt [µm]	Rp [µm]	Rv [µm]	Rq [µm]	Rc [µm]	Rsm [mm]	RVo [mm^3^/cm^2^]
S52	0.67 ± 0.06	4.87 ± 0.62	6.51 ± 2.05	2.65 ± 0.37	2.22 ± 0.28	0.85 ± 0.08	2.16 ± 0.17	0.053 ± 0.01	0.004621 ± 0.004444
S54	1.04 ± 0.05	6.81 ± 0.27	8.35 ± 0.62	3.67 ± 0.28	3.15 ± 0.12	1.3 ± 0.05	3.29 ± 0.15	0.064 ± 0.012	0.007899 ± 0.003347
S56	1.05 ± 0.07	7.34 ± 0.66	9.09 ± 1.52	3.99 ± 0.54	3.34 ± 0.28	1.35 ± 0.11	3.42 ± 0.24	0.06 ± 0.006	0.007015 ± 0.003543
S12	0.98 ± 0.08	6.74 ± 0.64	8.22 ± 0.82	3.58 ± 0.42	3.16 ± 0.27	1.24 ± 0.09	3.11 ± 0.32	0.052 ± 0.001	0.004303 ± 0.001137
S14	1.56 ± 0.06	10.92 ± 0.66	14.21 ± 1.9	6.05 ± 0.56	4.86 ± 0.34	1.99 ± 0.07	5.06 ± 0.32	0.073 ± 0.006	0.009353 ± 0.003343
S16	1.54 ± 0.13	10.01 ± 0.95	11.8 ± 1.07	5.23 ± 0.79	4.78 ± 0.45	1.93 ± 0.15	4.88 ± 0.22	0.082 ± 0.01	0.007461 ± 0.003409
S22	1.87 ± 0.1	12.71 ± 1.01	16.93 ± 4.8	6.42 ± 1.01	6.29 ± 0.58	2.39 ± 0.13	5.92 ± 0.28	0.091 ± 0.011	0.011335 ± 0.003755
S24	2.74 ± 0.15	17.72 ± 1.53	22.35 ± 3.37	8.55 ± 0.75	9.17 ± 0.96	3.46 ± 0.16	9.19 ± 0.77	0.108 ± 0.006	0.025713 ± 0.010871
S26	2.76 ± 0.12	17.35 ± 0.78	23.38 ± 2.11	8.32 ± 1.03	9.03 ± 0.36	3.46 ± 0.16	8.77 ± 1.1	0.117 ± 0.02	0.026591 ± 0.012697

**Table 5 materials-13-05824-t005:** The results for the material percentage in the profile.

Sample Number	Roughness Parameters				
	Rk [µm]	Rpk [µm]	Rvk [µm]	Mr1 [%]	Mr2 [%]
S52	2.18 ± 0.15	0.92 ± 0.18	0.83 ± 0.43	11.72 ± 1.05	90.28 ± 3.49
S54	3.42 ± 0.27	1.52 ± 0.29	1.37 ± 0.26	10.43 ± 2.21	88.68 ± 2.81
S56	3.32 ± 0.17	1.58 ± 0.2	1.28 ± 0.35	11.96 ± 1.95	89.62 ± 3.07
S12	3.10 ± 0.26	1.06 ± 0.18	0.91 ± 0.14	9.35 ± 1.33	90.65 ± 1.83
S14	4.93 ± 0.29	2.24 ± 0.3	1.76 ± 0.52	10.01 ± 1.24	89.44 ± 1.99
S16	4.99 ± 0.7	2.05 ± 0.46	1.63 ± 0.47	9.29 ± 3.78	91.17 ± 1.51
S22	5.92 ± 0.47	2.87 ± 0.47	2.27 ± 0.47	11.27 ± 1.12	90.16 ± 1.7
S24	8.76 ± 0.63	3.61 ± 0.69	3.98 ± 0.43	11.29 ± 2.46	87.26 ± 4.51
S26	9.24 ± 0.38	3.21 ± 0.94	4.08 ± 1.21	8.35 ± 1.22	87.53 ± 2.49

**Table 6 materials-13-05824-t006:** The percentage of stuck abrasive particles on the surface.

Sample Number	Percentage of Stuck Abrasive Particles [%]
S52	4.78 ± 1.16
S54	6.85 ± 0.64
S56	9.58 ± 1.76
S12	7.61 ± 0.88
S14	10.08 ± 1.26
S16	9.83 ± 1.4
S22	9.69 ± 1.48
S24	13.19 ± 2.13
S26	12.91 ± 3.04

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
