# Peer review of "The Surface Condition of Ni-Cr after SiC Abrasive Blasting for Applications in Ceramic Restorations"

_materials, 2020, doi:10.3390/ma13245824_

Round 1

Reviewer 1 Report

See the attached review comments.

Reviewer 2 Report

Journal: Materials

Ref manuscript ID materials-1008669

Manuscript title “The surface condition of Ni-Cr after SiC abrasive blasting for applications in ceramic restorations’’

Comments to authors:

This paper reads very comfortable. I can easily understand the idea and techniques. The motivation is clear to me. The results are comprehensive.

 In general, the article is good and is interesting, but it requires a major revision. There are some recommendations:

I recommend the authors when they revise their paper to make corrections and changes in the manuscript must be made in red color, and to facilitate the examiner for further consideration of it.

Therefore, the authors should revise the article by taking into consideration the following comments.

  • Nomenclature not specified anywhere in the manuscript (The document contains several unknown symbols and characters, and incomprehensible and so, a nomenclature is quite necessary for these components in order to describe them in order and in a clear and exhaustive manner. Some abbreviations like ; Ra, Rz, Rt, Rp, Rv, 86 Rq, Rc, Rsm, RV0 ,Rk, Rpk, Rvk, Mr1, Mr2 ,.. needs to be explained.  Please insert this nomenclature before the introduction section
  • In abstract the author must provide the highlight of this study.
  • The authors have dealt here with the industrial application of Abrasive blasting, therefore, the following definition should be added in the introduction section : “Abrasive blasting, more commonly known as sandblasting, is the operation of forcibly propelling a stream of abrasive material against a surface under high pressure to smooth a rough surface, roughen a smooth surface, shape a surface or remove surface contaminants”.
  • The authors said this “The purpose of this paper was the analysis of the impact of the parameters of the abrasive blasting on the condition of the surface of the Ni-Cr alloy” while Nickel-Chrome (Ni-Cr) Alloys properties and his applications, advantages and disadvantages were not cited in the document, no information was provided
  • Figure 1 is not cited in the manuscript, why , in addition it must be interpreted physically in a very intense . please , add a reference to this figure
  • Authors should improve the English of this paper, in order to enable readers to understand the content of your paper easily; you should check your paper carefully and revise the grammar mistakes. Indeed, there are many errors in this document, some words are missing in several places, punctuation was not observed along the paper also lacks a comma (,) and point in several places, some sentences have meaning disorders and should be checked. Unsatisfied spacing between the words and the quotation of the units in several places, we can cite by way of example only :

- In page 1, Abstract ,  line 15 , comma(,) is missing , "  wettability and percentage " should be corrected to "  wettability, and percentage….”

section introduction

- In page 1, line 30-31, "   by machining in  a small degree but abrasive

" should be corrected to "   by machining to  a small degree, but abrasive”

- In page 1, line 34 , "the type of the abrasive " should be corrected to "  the type of abrasive”

- In page 1, line 35 , comma(,) is missing "  size and shape and, finally they can " should be corrected to "    size, and shape, and, finally, they can”

- In page 1, line 37 , "  influence " should be corrected to "  influences”

- In page 2, line 55 , comma(,) is missing" cobalt-chromium alloy or titanium " should be corrected to " cobalt-chromium alloy, or titanium”

- In page 2, line 62 , " to analyze to the condition " should be corrected to " to analyze the condition”

section 2

- In page 2, line 77, " deionised " should be corrected to " deionized”

- In page 2, line 84 , " The length of the profile " should be corrected to " The length of profile”

- In page 3, line 88, , " The received " should be corrected to " Received”

- In page 3, line 95, , " between places " should be corrected to " between the places”

section 3

- In page 4, line 108, " The greatest differences " should be corrected to " The great differences”

- In page 6, line 154, , " with an increased " should be corrected to " with an increase”

section 4

- In page 7, line 177, " using a smaller " should be corrected to " using smaller”

- In page 7, line 182, " at a high pressure, " should be corrected to " at high pressure,”

- In page 7, line 206, " were observed for the surface after the processing " should be corrected to " were observed on the surface after processing”

- In page 8, line 210, comma(,) is missing" was used but " should be corrected to " was used, but”

- In page 8, line 228 " Also, the surfaces " should be corrected to " Also, surfaces”

- In page 8, line 237 " The uneven " should be corrected to " Uneven”

- In page 8, line 243 " with a growing " should be corrected to " with growing”

- In page 8, line 255 " is a less hard " should be corrected to " is less hard”

- In page 9, line 270 " appropriate but " should be corrected to " appropriate, but”

- In page 9, line 270 " the connection hence " should be corrected to " the connection, hence”

Please, there are many other errors in this document in several places, please carefully refine the English language.

  • Use of articles 'The', 'A' and 'An' needs to be corrected at some places
  • The results of surface roughness measurement have been grouped by the authors in tables 2 and 3. but the authors did not say how they got these resultants.

I recommend here the authors to clearly indicate the detailed experimental setup of the research for the measurement of the surface roughness, please add an illustrative diagram or figure to clarify these results

  • The introduction section is far failed to express the aim of this study and has to be reworked. It is crucial to answer the following questions:
       What is new in this study?
                       b.     What do you show in this study?
                       c.     How do those previous research works impact your work?

  • Authors should explain clearly more about the novelty of their work in introduction
  • A simple flow chart is required to represent the steps-wise procedure followed for carrying out the analysis.
  • Key assumptions and their implications could have been elaborated
  • All the thermophysical and characteristics of the material model used in the analysis must be provided as tables.
  • The quality of the figures in this document needs to be improved; the figures need to be larger in size so the data and labels can be clearly read.
  • The introduction should provide a clear statement of the problem, the relevant literature on the subject, and the proposed approach or solution. It is be understandable to colleagues from a broad range of scientific disciplines. For that purpose I would like request author to add the following recent papers, which can help to enhance the introduction section:

  • Effects of Material Properties on Generation of Brake Squeal Noise using Finite Element Method, Latin American Journal of Solids and Structures 12(8):1432-1447

  • The authors must provide a greater discussion of the results.
  • In 'Result and Discussion' authors have noted observations. But it is suggested that to provide physical explanations of all obtained results which can enrich the quality of the paper.
  • The present results of this manuscript does not compare with any previous results. It is highly recommended to compare the present results with any other previous study as they mentioned in the abstract about "reexamined".
  • It seems to me that the conclusion section is very short and at the same time poor requires revision. The conclusion should be clear. In conclusions please provide a general comment on the obtained results. For example: how can this work help in future researchers to contribute further knowledge?

Altogether, the paper needs modification to be suitable for the standards required for publication; therefore I recommend that it required to major revision.

I look forward to receiving the revised version of this manuscript

My best regards

Round 2

Reviewer 2 Report

Journal:  Materials

Ref manuscript ID materials-1008669

Manuscript title “The surface condition of Ni-Cr after SiC abrasive  blasting for applications in ceramic restorations”

Dear  
Ms. Andreea Curea

Assistant Editor

MDPI Open Access Publishing Romania SRL

I have completed the review process of the revised version of the above-mentioned document, and after careful reading and focused on it, and after careful checking of all the changes inserted in the document, accuracy of the figures, tables ; nomenclature and citation references, I strongly believe that the authors have swept all the remarks and the points that I raised them, and so, they responded positively on all my comments.

In general, the entire article seems to provide compelling answers to all my concerns and deserves to be published in this journal, I recommend the publication of the article as it stands, in its fully revised version.

The authors made all required corrections.
The paper now is accepted without any changes.

Thank you again for giving me a chance and inviting me to review this document.

Best regards
